# Communities of Uropodina (Acari: Mesostigmata) in Nest Boxes Inhabited by Dormice (*Glis glis* and *Muscardinus avellanarius*) and Differences in Percentages of Nidicoles in Nests of Various Hosts

**DOI:** 10.3390/ani13223567

**Published:** 2023-11-18

**Authors:** Jerzy Błoszyk, Grzegorz Hebda, Marta Kulczak, Michał Zacharyasiewicz, Tomasz Rutkowski, Agnieszka Napierała

**Affiliations:** 1Department of General Zoology, Faculty of Biology, Adam Mickiewicz University, Uniwersytetu Poznańskiego 6, 61-614 Poznań, Poland; bloszyk@amu.edu.pl (J.B.); markul12@amu.edu.pl (M.K.); zacharyasiewicz@gmail.com (M.Z.); tomasz.rutkowski@amu.edu.pl (T.R.); 2Natural History Collections, Faculty of Biology, Adam Mickiewicz University, Uniwersytetu Poznańskiego 6, 61-614 Poznań, Poland; 3Institute of Biology, Opole University, Oleska 22, 45-040 Opole, Poland; grzesio@uni.opole.pl

**Keywords:** *Apionoseius infirmus*, bird nests, dormice, *Leiodinychus orbicularis*, mammal nests, merocenose, nidicoles

## Abstract

**Simple Summary:**

Hanging nest boxes, which are used by various groups of animals such as birds or mammals (e.g., dormice and bats), increase the number of shelters and breeding places for these often rare animals. Nest boxes not only become habitats for the host, but are also inhabited by various groups of invertebrates, including insects, spiders, millipedes, snails, and also small arachnids, which are mites. In this article, we present an analysis of the community of one of the groups of mites—Uropodina—which also inhabit nest boxes. In the examined boxes, five species belonging to the discussed group were found, out of which only one (*Leiodinychus orbicularis*) is a nidicole, i.e., a species that inhabits the nests of various animals. This article also analyses the habitat preferences of the mentioned species and another Uropodina species associated with nests—*Apionoseius infirmus*. It was proven that *L. orbicularis* clearly dominated both in the examined dormouse, bat, and bird boxes, whereas *A. infirmus*, which was less numerous in the communities, preferred natural nests, including the nests of birds of prey. The clear dominance of *L. orbicularis* in the examined boxes can be explained by the specific microclimate, such as very low humidity, which prevails in the boxes.

**Abstract:**

Bird and mammal nests and nest boxes constitute microenvironments in which various groups of invertebrates can live, including mites from the suborder Uropodina (Acari: Mesostigmata). The main aim of the current study was to ascertain the characteristics of mite communities from the suborder Uropodina, which inhabit the nests of dormice (Gliridae) built in nest boxes. The second aim of the study was to compare the habitat preferences of *Leiodinychus orbicularis* (C. L. Koch) and *Apionoseius infirmus* (Berlese), i.e., two typically nest-dwelling species of Uropodina. The material for the study was collected from nest boxes in six forest complexes in southwestern Poland. The conducted research revealed the presence of five species of Uropodina, with a total number of 559 specimens, in the examined boxes. *Leiodinychus orbicularis* was found in almost half of all of the examined boxes and was a superdominant species in the communities. The analysis of the habitat preferences of the two nest species of Uropodina showed that *A. infirmus* preferred old natural nests, in which the communities were formed from a larger number of species, without a significant statistical prevalence of one species. On the other hand, *L. orbicularis* occurred sporadically in open bird nests, but was very numerous and frequent in nest boxes. The significant dominance of *L. orbicularis* in nest boxes can probably be explained by the specific conditions prevailing in this type of microhabitat, including the very low humidity and food resources that this mite species prefers compared to other species of Uropodina.

## 1. Introduction

“Ecological niche” is a term for the position of a species within an ecosystem, describing both the range of conditions necessary for the persistence of the species and its ecological role in the ecosystem [1]. Every species strives to maximise the use of the available niches and populate them with individuals. That is why the number of inhabited environments or microenvironments and the number of local populations can be considered as indicators of a species’ evolutionary success.

The nesting abilities of secondary cavity nesters, bats, and Gliridae mammals depend on the presence of natural cavities that are necessary for establishing nests [2,3,4,5]. The availability of nesting sites for species inhabiting natural cavities is limited, especially in younger commercial forests. The number of nesting sites is regularly increased by creating “artificial cavities”, that is, by hanging nest boxes for particular groups of animals (e.g., for birds [2,3,4,5]; for dormice [6,7]; and for bats [8]). By creating artificial shelters and breeding places for birds and some endangered mammals, humans contribute to the creation of new niches for many invertebrate species. Bird nest boxes, bat boxes, and less commonly encountered boxes intended for mammals from the Gliridae family are specific types of microenvironments (merocenose) inhabited by diverse groups of invertebrates including Arachnida and Insecta (esp. species from orders such as Coleoptera, Diptera, Siphonaptera, Hemiptera, Hymenoptera, and Lepidoptera), and even invertebrates that are not typically associated with nest boxes on trees such as Isopoda, Gastropoda, and Myriapoda [9,10,11,12,13,14,15,16]. Much scholarly attention was paid to the presence of ectoparasites in such places related to their hosts (fleas, ticks, Diptera: Protocalliphoridae, and some mites) [17,18,19,20,21,22,23,24]. Among the groups that are frequently observed in nest boxes, the most interesting phenomena is the presence of typical nidicoles, for whom nests are the proper type of habitat [25,26].

Nest boxes may contain different materials of organic origin. Typically, their most considerable portion consists of nest material, which may be composed of both plant (stems, leaves, the roots of plants, and mosses) and animal components (feathers, fur, and hairs), and other remains left by the host, including faeces, pellets, food storages, and remnants after broods (egg shells and dead juveniles) [27]. Such diverse nest box contents can attract organisms presenting different foraging strategies. These organisms are saprophagous species that feed on decomposing nest materials or bird and mammal prey and dropping remains [28,29]; scavengers and carnivores, which feed on all developmental stages of other invertebrates commonly living in the nest [16,26,30]; and vertebrate ectoparasites, which spend at least part of their lives buried in nest material [18,31]. Apart from the direct trophic benefits for organisms inhabiting nest boxes, other species can also be associated with more favourable conditions occurring in nest boxes than in natural conditions, for example, some groups of hymenopterans like ants, bumblebees, and social wasps [14,32].

Previous studies have shown that the microhabitats of bird and mammal nests are often also inhabited by mites from the suborder Uropodina (Acari: Mesostigmata). Uropodina mites were found both in the nests and nest boxes of various bird species [25,30,33,34,35,36,37,38,39,40,41,42,43], mole nests and badger burrows [44,45,46], and bat boxes [47]. The results obtained so far have shown that nests constitute various environments for Uropodina. The community structure of Uropodina in these microhabitats depends on different factors, such as the nesting host ecology, the duration of the nest existence, and the location of the nest. As far as the time of the nest’s existence is concerned, communities of Uropodina have been examined so far in nest boxes [34,47], one-year natural nests [39,43], and perennial nests of birds of prey [36,38,41,48,49,50,51,52,53]. Other important microhabitats for Uropodina communities are perennial nests of mammals, such as burrows of small and medium mammals (including mole (*Talpa europaea* L.), marmot (*Marmota marmota latirostris* Kratochvíl), and badger (*Meles meles* L.)) [45,46]. These nests, especially badger burrows, can exist for a very long time [54], which enables the formation of diverse Uropodina communities [45]. The factor of nest existence is also very important because of the slow rate of colonisation observed in Uropodina species in this type of unstable microhabitat. The most frequent method of colonising nests as well as other types of microhabitats used by Uropodina mites is phoresy [55].

It is also worth mentioning that most of the research conducted so far has focused mainly on Uropodina communities inhabiting arboreal or aboveground bird nests [36,39,48,49,50,51,52,53,56,57]. However, recent studies on Uropodina communities found in nests of the wood warbler (*Phylloscopus sibilatrix* (Bechstein)), the passerine species, which builds its nests on the ground, have revealed that there is a lack of typical nidicoles in such nests and that the community structure is very similar to those found in the soil and nests of the common mole [35,46] compared to those recorded in other nests of birds. 

The Uropodina species, which is most frequently and abundantly found in the nests of various bird and mammal species, is *Leiodinychus orbicularis* [55], described by Koch in 1839. The typical nidicoles associated with mammal nests are *Phaulodiaspis rackei* (Oudemans), *Ph. advena* (Trägårdh), and *Ph. borealis* (Sellnick), which inhabit underground nests of the mole, the marmot, and the European water vole (*Arvicola amphibius* L.) [35]. In nests of various bird species, *Apionoseius infirmus* (Berlese) is often and numerously found, while *Nenteria pandioni* Wiśniewski *et* Hirschmann occurs exclusively in the nests of the white-tailed eagle (*Haliaeetus albicilla* L.). Nest boxes inhabited by dormouse mammals have not yet been studied for the presence of Uropodina mites.

Dormice (Gliridae) are a family of mammals from the suborder Sciuromorpha in the order Rodentia. In Poland, they are represented by four species, the European edible dormouse (*Glis glis* L.), the garden dormouse (*Eliomys quercinus* L.), the forest dormouse (*Dryomys nitedula* (Pallas)), and the hazel dormouse (*Muscardinus avellanarius* L.), all of which are legally protected, with some still requiring active protection. The presence of dormice is associated with the presence of deciduous and mixed forests of high natural value, with an availability of trees with hollows, in which they shelter, reproduce, and store food [58]. The lack of old deciduous tree stands with numerous trees with hollows, which are also the natural habitats of dormice, creates, like in the case of birds, the need to hang special boxes that serve as their substitute shelters. Indeed, dormice readily occupy nest boxes, and providing these artificial shelters has become the basic method in studies of many aspects of dormice biology [6,7,59,60,61] and the impact of Gliridae on hole-nesting birds [62,63]. For this reason, we decided to study the communities of Uropodina mites inhabiting nest boxes occupied by dormice.

The collection of material from several nest boxes inhabited by dormice in southwestern Poland allowed us, for the first time, to characterise the communities of Uropodina inhabiting these nest boxes. That is why the aim of this study was to ascertain the characteristics of mite communities from the suborder Uropodina, which inhabit the nests of dormice (Gliridae) built in nest boxes. In addition, an analysis of the occurrence and ratio of two nest species of Uropodina, i.e., *L. orbicularis* and *A. infirmus*, in the examined nests of different hosts was also carried out. 

## 2. Materials and Methods

### 2.1. Study Area

This study was conducted in six forest complexes in the central and southern parts of Opolskie voivodeship (south and southwestern Poland). Two sites are located in the Opawskie Mountains, and the four other sites are located in the lowland part of the Opole region, that is, in the Stobrawski Landscape Park and Niemodlin Forest (see characteristics: Table 1 and Figure 1A). Forest complexes were predominantly deciduous and mixed old forests, with multiple horizontal layers and beech and oak as the dominant tree species, which makes them attractive for dormice. In each of the four lowland forest complexes, groups of 24 designed dormice nest boxes were provided in 2020. In the two mountain forests, only bird nest boxes were present.

### 2.2. Data Collection

The samples were collected from wooden nest boxes, both those designed for dormice and those designed for birds (for nest boxes, see Table 1). All dormice and bird nest boxes were checked yearly since 2020 and cleaned before the next season. The selected nest boxes were examined once between the 12th of October 2022 and the 19th of January 2023. 

In this study, we analysed material from only 38 nest boxes where typical remnants of dormice presence were left: 15 nest boxes designed for dormice (Figure 1B–D) and 23 typical bird nest boxes (Table 1). The dimensions of the examined dormice boxes were as follows: diameter of opening: 4.5 cm, bottom: 14 × 16 cm, distance from the opening to the bottom: 25 cm, distance from the top to the bottom: 33 cm. The openings of the dormice boxes were facing the tree trunk, and the box was fixed to the tree at a distance of 4.5 cm from the wooden pole. The dimensions of the bird boxes were more variable, and they were as follows: diameter of opening: 3.5–4.5 cm, bottom: 14 × 14–16 cm, distance from the opening to the bottom: 17–25 cm, distance from the top to the bottom: c. 25–35 cm. The dormice and bird nest boxes were placed on trees, c. 4–5 m above the ground, and only such nest boxes were examined, where typical remnants of edible or hazel dormouse were left, including nests, aggregations of leaves, droppings, gnawed nuts of hazels, or beech and oak seeds. 

During the box examination, its rough qualitative content characteristics were registered, including the presence of bird or mammal nest remains, faeces, or food storage. The entire contents of the examined nest boxes were placed into sealed plastic bags with labels describing the box’s location and date of collection. 

The samples were then immediately transferred to Berlese -Tullgren funnels for mite extraction. This process lasted 72–96 h for each sample, depending on its volume. The extracted specimens were collected in Eppendorf tubes filled with c. 70–80% ethanol alcohol and labelled. The mite specimens were sorted and identified with a stereoscopic microscope (Olympus SZX 16), and the identification of the extracted species was conducted by the first author on the basis of the publications by Karg [64], Błoszyk [65], and Mašán [66]. The extracted specimens were stored in the Natural History Collections (Faculty of Biology) at Adam Mickiewicz University in Poznań.

### 2.3. Data Analysis

The structures of the analysed mite communities are characterised with the index of dominance (D) and the frequency of occurrence (F). The scale has the following classes: dominance D5 eudominants (>30.0%), D4 dominants (15.1–30.0%), D3 subdominants (7.1–15.0%), D2 recedents (3.0–7.0%), and D1 subrecedents (<3.0%); frequency F5 euconstants (>50.0%), F4 constants (30.1–50.0%), F3 subconstants (15.1–30.0%), F2 accessory species (5.0–15.0%), and F1 accidents (<5.0%) [43]. The average number of specimens in positive samples, presented in Table 2, includes only the nest boxes occupied by dormice. The data used to compare the occurrences of *L. orbicularis* and *A. infirmus* in nests of different hosts (Table 3) were stored in the computer database in the Natural History Collections (Faculty of Biology).

### 2.4. Statistical Analysis

We used non-parametric tests (Fisher’s exact test and Mann–Whitney U test). The established significance level in the statistical analysis was *p* < 0.05. All probability values shown here are two-tailed. All statistical analyses followed the formulae in STATISTICA 12.0 [67].

## 3. Results

### 3.1. Characteristics of Uropodina Communities in Nests of Species from Gliridae Family

Mites from the Uropodina group were present in the nest boxes in all six forest complexes, and 50% of the examined boxes for the purpose of this study contained at least one specimen of Uropodina (Table 2). In the 38 nest boxes, the presence of five species of Uropodina was recorded. *Leiodinychus orbicularis* turned out to be the most numerous species. A total of 559 specimens of this species were found, including 184 females, 153 males, 165 deutonymphs, and 57 protonymphs. Moreover, *L. orbicularis* was the superdominant species in the examined community and was present in nearly half of the examined nest boxes. It was present both in bird nest boxes (11 boxes) and dormice nest boxes (6); there were no differences in the type of box selection (Fisher’s exact test, *p* = 0.56). The mean number of *L. orbicularis* also did not differ between the types of nest boxes (Mann–Whitney U test; U = 23.0, *p* = 0.48). In individual boxes, the presence of 1 to 285 specimens was recorded (on average, in one nest, there were 29.7 specimens ± 65.8) (Table 2). 

### 3.2. Frequency and Abundance of Nidicoles Leiodinychus Orbicularis and Apionoseius Infirmus in Material from Nests of Different Hosts

The analysis of the species composition of the examined Uropodina communities found in nests and nest boxes inhabited by various species of mammals and birds showed that *L. orbicularis* occurs in most of the merocenoses of this type that have been analysed so far (Table 3). However, the percentage of this species in Uropodina communities in such microhabitats varies (Table 3). A very high percentage of this species (even >90%) was recorded in boxes for birds, bats, and dormouse mammals. These are the communities with a small number of species, which means that *L. orbicularis* is a superdominant species in such cases. On the other hand, *A. infirmus* has not been found so far in most nests found in boxes; it only occurred in boxes inhabited by starlings, but the percentage of this species and the frequency were small there. This species also did not occur in the nests of thrushes, the nests of wood warblers, and in underground mole nests.

## 4. Discussion

In the examined boxes inhabited by dormice, the occurrence of five species of Uropodina was recorded, of which only one, i.e., *L. orbicularis*, can be typically considered as nest species [25,34,55]. The other species that were found there, such as two species from genus *Trachytes*, and species from genus *Nenteria*, such as *N. splendida*, were soil species, associated with the litter and soils of different forests or open environments [65]. Occasional adult specimens (lack of juvenile forms) probably found themselves there accidentally with the nesting material or food collected from the ground. Previous studies have shown that *L. orbicularis* is a nidicole associated with various types of bird nests, mammal nests, and nest boxes [25,34]. The boxes inhabited by dormice are another microhabitat, in which the presence of juvenile forms of this species, especially protonymphs, shows that *L. orbicularis* can live and reproduce in such places. The research also did not reveal any differences in the preferences of *L. orbicularis* in relation to the type of nest box host (bird vs. dormice). This means that the specific microclimate in the boxes is the factor that attracts this mite species.

The comparison of Uropodina communities from the nests of other birds and mammals showed that in artificial, human-made microhabitats, such as bird nest boxes, dormouse boxes, and bat boxes, this species has the largest percentage in the whole community (even above 90%). However, these are usually communities with a low number of species, in which *L. orbicularis* monopolises all available resources (Table 3). In typical open bird nests, it occurs rather sporadically. The only exception in this respect are nests of the white stork, in which it was found to be relatively numerous and frequent [39]. It seems that this species avoids the nests of sparrows located on the ground (as seen in the lack of the species in nests of wood warblers [43], mammal burrows (mole and badger), and tree nests of birds of prey [39]) (see Table 3). 

The absence of this species in the nests of birds of prey (e.g., eagles and the white-tailed eagle (*Haliaeetus albicilla* (L.))), ground nests of the wood warbler, and underground burrows of the badger, and the low percentage of the specimens in other underground mammal nests is probably due to the method of dispersion of the species, namely the deutonymphs of *L. orbicularis* (with pedicels), which are found in nests and are carried by insects [41]. However, no carrier species have been identified yet, though it is assumed that they are probably carried by beetles (unpublished data). The peculiar habitat preferences of *L. orbicularis* determine the preferred type of the nests inhabited by this species, excluding those located underground. It is worth mentioning that the discussed species is characterised by a wide ecological valence, which allows it to occupy various niches, including those created by humans, and for this reason, it was also found in stored products [68]. It is possible that the numerous occurrences of nests built in nest boxes are related to the possibility of colonising environments of anthropogenic origin.

The second of the analysed nidicoles, i.e., *A. infirmus*, was not found in the analysed material from the boxes inhabited by dormice. As for the boxes, the species was only found in the nests of the starling (Table 3). Besides this, it occurred mainly in the nests of birds of prey, in the nests of kites, and in the nests of both species of the stork. In the nests of the black stork, black kite (*Milvus migrans* (Boddaert)), and red kite (*Milvus milvus* (L.)), the abundance of this species was higher than that of *L. orbicularis*. Generally, it can be stated that unlike *L. orbicularis*, *A. infirmus* avoids nests built in nest boxes, but it is more often present in old natural nests, where there are usually more species in the community, without a clear statistical prevalence of one species.

## 5. Conclusions

Apparently, little is known about the method of dispersion of nidicoles and the routes by which they reach isolated microhabitats, such as nest boxes. The presence of phoretic deutonymphs suggests that the discussed species of Uropodina are carried by insects, probably beetles (like most phoretic Uropodina). However, specific vector species have not been found yet. The clear dominance of *L. orbicularis* in the examined boxes (regardless of the host that inhabited them) can be explained by the specific microclimate that prevails in the boxes [69]. This species tolerates very low humidity, which is seen in most boxes, better than other Uropodina species. Finally, it cannot be ruled out that under such conditions in the boxes, or more precisely, in the nesting material, specific fungi can grow, which are probably the food of this mite species.

## Figures and Tables

**Figure 1 animals-13-03567-f001:**
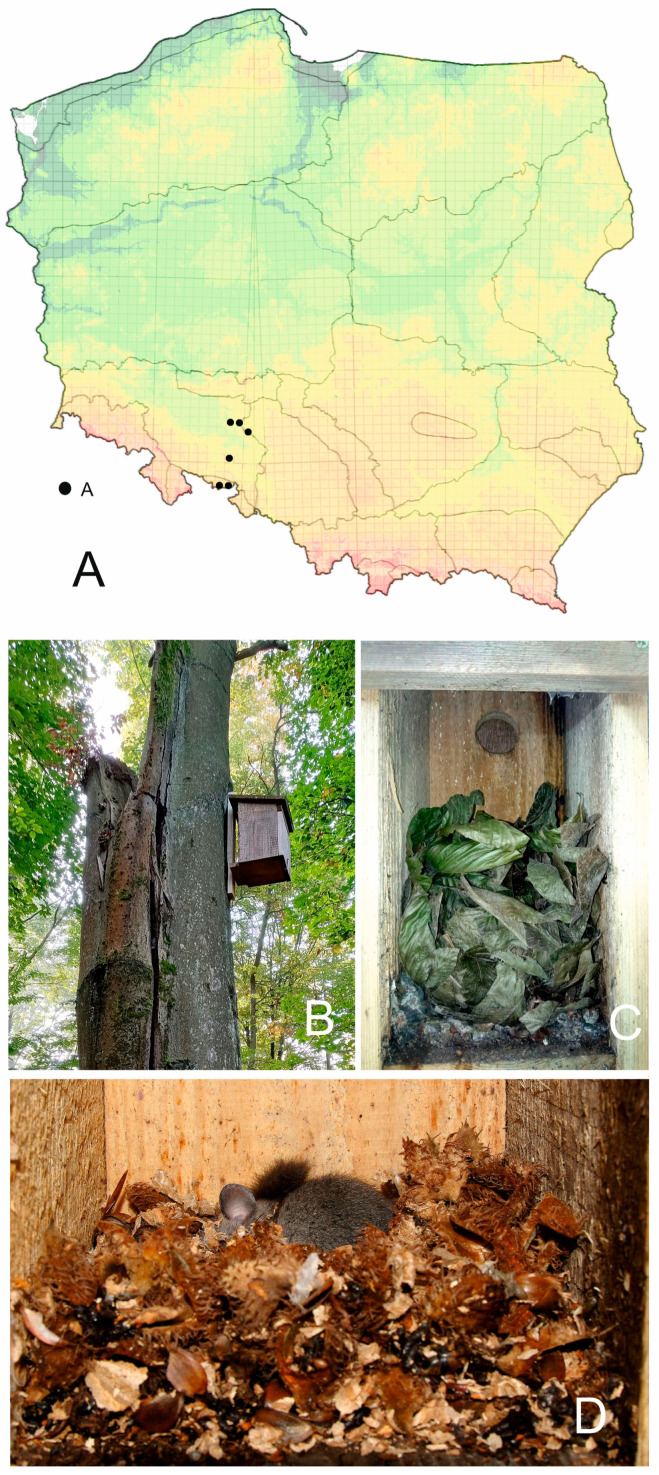
(**A**) Location of the study sites in Poland (black dots) (UTM 10 × 10 km). (**B**) Typical designed dormice nest box placed on beech; the opening is facing the tree trunk. (**C**) Designed dormice nest box with a bulk of leaves and droppings of European edible dormouse. (**D**) Designed dormice nest box with sleeping European edible dormouse and gnawed beech seeds.

**Table 1 animals-13-03567-t001:** Characteristics of six study sites with nest boxes; number of examined nest boxes in brackets.

Study Site; GPS of Central Point of the Study Site	Habitat, Dominated Tree Species; Type of Nest Boxes	Dormice Species: Nest Box Content during Sampling
Opawskie Mts: Pokrzywna	Deciduous forest, beech; bird nest boxes (19)	Edible dormice: leaves of trees, gnawed beech and oak seeds, droppings
GPS: 50.2780, 17.4514		
Opawskie Mts: Dębowiec	Deciduous forest, beech; bird nest boxes (4)	Edible dormice: leaves of trees, gnawed oak seeds, droppings
GPS: 50.2787, 17.5398		
Stobrawski Landscape Park: Kup	Mixed forest, pine, and beech; dormice nest boxes (6)	Edible dormice: leaves of trees, gnawed beech seeds, droppings
GPS: 50.8613, 17.9292		
Stobrawski Landscape Park: Lubsza	Deciduous forest, beech, oak; dormice nest boxes (4)	Hazel dormice: nests
GPS: 50.9328, 17.5658		
Stobrawski Landscape Park: Kozuby	Deciduous forest, oak; dormice nest boxes (3)	Edible and hazel dormice: nests, gnawed oak and beech seeds, leaves of trees, droppings
GPS: 50.9381, 17.8124		
“Niemodlin Forest”: Goszczowice	Deciduous forest, beech, dormice nest boxes (2)	Edible dormice: leaves of trees, gnawed beech seeds, droppings
GPS: 50.5784, 17.6084		

**Table 2 animals-13-03567-t002:** Species composition, number of specimens (N), dominance (%), frequency (F%), and average number of specimens in positive samples from nests of dormice. SD—standard deviation; F—females; M—males; D—deutonymphs; P—protonymphs; L—larvae.

Species	N	D%	F%	Average ± SD	Max.	F	M	D	P	L
*Leiodinychus orbicularis* (C.L. Koch)	559	99.1	46.0	32.9 ± 69.1	285	184	153	165	57	-
*Trachytes aegrota* (C.L. Koch)	2	0.5	2.7	2.0	2	2	-	-	-	-
*T. irenae* Pecina	1	0.2	2.7	1.0	1	1	-	-	-	-
*Neodiscopoma splendida* (Kramer)	1	0.2	2.7	1.0	1	1	-	-	-	-
*Nenteria* sp.	1	0.2	2.7	1.0	1	-	1	-	-	-
Total	564	100.0	50.0	29.7 ± 65.8	285					

**Table 3 animals-13-03567-t003:** Occurrence of *L. orbicularis* and *A. infirmus* in nests of different hosts: A—bat boxes; B—nest boxes occupied by dormice (*Glis glis* (L.) and *Muscardinus avellanarius* (L.)); C—nests of tits (Paridae sp.) and flycatchers (*Muscicapa* sp.) in boxes; D—nests of starlings (Sturnidae sp.) in boxes; E—nests of white storks (*Ciconia ciconia* (L.)); F—nests of thrushes (Turdinae sp.); G—nests of black storks (*Ciconia nigra* (L.)); H—nests of kites (*Milvus* sp.); I—burrows of various mammals; J—mole nests (*Talpa europaea* L.); K—tree nests of various birds of prey; L—wood warbler nests (*Phylloscopus sibilatrix* (Bechstein)); M—badger (*Meles meles* (L.)) burrows. Bold—highest dominance and frequency in examined communities.

	A	B	C	D	E	F	G	H	I	J	K	L	M
Number of boxes or nests	58	38	170	103	38	47	39	52	23	132	34	66	32
Number of Uropodina species	2	5	3	2	11	15	11	11	24	15	11	14	16
Number of specimens	119	564	453	1525	2827	275	373	942	782	4718	925	595	413
*L. orbicularis*													
Number of specimens	118	559	443	1281	904	42	49	7	2	5	0	0	0
Dominance (%)	**99**	**99**	**98**	**84**	32	15	13	>1	>1	>1			
Frequency (%)	19	46	11	21	**74**	6	5	4	4	>1			
Average number of specimens in a nest ± SD	10 ± 14	32.9 ± 69.1	23 ± 52	58 ± 151	32 ± 91	14 ± 14	24 ± 16	0.1 ± 0.7	2	5			
*A. infirmus*													
Number of specimens	0	0	0	244	26	0	270	225	1	0	289	0	4
Dominance (%)				16	>1		21	24	>1		10		1
Frequency %				10	**26**		**31**	18	4		**32**		9
Average number of specimens in a nest ± SD				24 ± 28	2 ± 2		22 ± 141	22 ± 140	1		26 ± 53		1

## Data Availability

The data presented in this study are stored in an Invertebrate Fauna Bank (Natural History Collections, Faculty of Biology, Adam Mickiewicz University, Poznań, Poland).

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
