# Peer review of "Communities of Uropodina (Acari: Mesostigmata) in Nest Boxes Inhabited by Dormice (Glis glis and Muscardinus avellanarius) and Differences in Percentages of Nidicoles in Nests of Various Hosts"

_animals, 2023, doi:10.3390/ani13223567_

Round 1

Reviewer 1 Report

Comments and Suggestions for Authors

Manuscript contains interesting analysis of uropodine mite fauna inhabiting both the bird- and dormice-boxes. I assume, that the results (with superdominance of L. orbicularis) encouraged the Authors to compare present data with its presence within other habitats, including also analysis of the other species inhabiting various nests i.e. A. infirmus. In Table 2. the Authors could in my opinion present the data for bird-boxes and dormice-boxes separately. With regard to the results: could the Authors explain why there were no larvae of L. orbicularis found in such a numerous population? In Table 3. I suggest to add a top-row with explanation of the nest type (box nests vs. natural nests). The main problem with data in the manuscript is that in Table 3. the Authors included results from other papers. I strongly recommend to add a bottom-row in Tab. 3. with clearly indicated present results (column B) and references for the other columns. Table 3. is cited in both the results and discussion and there should be a clear statement about which part of this table is an original result and which part is used to discuss it. In the line 223-224 the Authors omitted the absence of A. infirmus in nests of the wood warbler.

Author Response

Reviewer #1

Communities of Uropodina (Acari: Mesostigmata) in nest boxes inhabited by dormice (Glis glis and Muscardinus avellanarius) and differences in percentage of nidicoles in nest of various hosts

The authors of the study are grateful to the reviewer for all comments and suggestions. Most of them have turned out to be extremely helpful, which obviously has considerably improved the overall quality of the manuscript. The more detailed comments to the remarks are as follows: 

Detailed responses to the Reviewers comments: 

Response to Reviewer #1:

COMMENTS TO THE AUTHORS:

In Table 2. the Authors could in my opinion present the data for bird-boxes and dormice-boxes separately.

  • We decided that breaking down this table into data from bird boxes occupied by dormice and boxes hung for dormice occupied by dormice does not make sense, due to the too small number of samples. Besides, both types of boxes were very similar in terms of dimensions and construction (please, see added pictures), the only difference was the location of the entrance hole

With regard to the results: could the Authors explain why there were no larvae of L. orbicularis found in such a numerous population?

  • The lack of larvae at the periods when the samples were collected is due to the phenology of the species. A large number of larvae occur in nests in October.

In Table 3. I suggest to add a top-row with explanation of the nest type (box nests vs. natural nests). The main problem with data in the manuscript is that in Table 3. the Authors included results from other papers. I strongly recommend to add a bottom-row in Tab. 3. with clearly indicated present results (column B) and references for the other columns. Table 3. is cited in both the results and discussion and there should be a clear statement about which part of this table is an original result and which part is used to discuss it.

  • In the caption of Table 3, there is information about which data come from boxes and which from natural nests, therefore adding an additional row in the table would duplicate the existing information.

As for the data included in this table used for comparisons, these are data from the computer database in the Natural History Collections, not all of which have been published yet. Information about where the data used in this table come from has been added in subsection 2.3. Data analysis.

In the line 223-224 the Authors omitted the absence of A. infirmus in nests of the wood warbler.

  • This information has been added.

Reviewer 2 Report

Comments and Suggestions for Authors

Dear Authors,

It is an interesting study but writing requires improvements. I include some comments in the text. Some sentences are unclear and require rephrasing. Also some  issues regarding the methods should be clarified. 

Author Response

Reviewer #2

Communities of Uropodina (Acari: Mesostigmata) in nest boxes inhabited by dormice (Glis glis and Muscardinus avellanarius) and differences in percentage of nidicoles in nest of various hosts

The authors of the study are grateful to the reviewer for all comments and suggestions. Most of them have turned out to be extremely helpful, which obviously has considerably improved the overall quality of the manuscript. The more detailed comments to the remarks are as follows: 

Response to Reviewer #2:

We thank you for all the comments included in the text - we have taken almost all of them into account and it has certainly improved the quality of the manuscript.

We wanted to clarify that the positive samples that were analysed in Table 2 are samples in which Uropodina specimens were found. However, the data used for the analyses in Table 3 came from a computer database in the Natural History Collections. This information has been added in subsection 2.3. Data analysis.

In the text, however, we did not use square brackets, as suggested by the reviewer, because they are used for citing literature.

Reviewer 3 Report

Comments and Suggestions for Authors

Enclosed as pdf

Author Response

Reviewer #3

Communities of Uropodina (Acari: Mesostigmata) in nest boxes inhabited by dormice (Glis glis and Muscardinus avellanarius) and differences in percentage of nidicoles in nest of various hosts

The authors of the study are grateful to the reviewer for all comments and suggestions. Most of them have turned out to be extremely helpful, which obviously has considerably improved the overall quality of the manuscript. The more detailed comments to the remarks are as follows: 

Detailed responses to the Reviewers comments: 

Response to Reviewer #3:

COMMENTS TO THE AUTHORS:

Comments on the MS: 1. The authors gave the main aim of investigation in the Abstract, but did not mention about it in the end of Introduction or before the Methods.

 - The aim of the study is now explained in the end of Introduction.

  1. In the abstract the authors wrote about significant dominance of L. orbicularis in the nest boxes, but did not mention clearly about it in the Results (text or Tables).

-  The section Results explains that L. orbicularis in the nest boxes was the most abundant species (superdominant), and for this reason we think that its role in the described community was adequately emphasized.

  1. The References need some checking and improvements: - the reference “1. Polechová, J.; Storch, D. 2008. Encyclopedia of Ecology. Volume 3. 2019; pp. 72-80.” need changing according to the requirements of the Editor; - the authors use between number of pages short (-) and long dash (–), and wrote volume in italic or normal letter – example 2016, 18(1), 279-299.; 2022, 15(4), 461–469. – please check all references and change them according to the requirements of the Editor.

- The mistakes in the section References have been corrected.

Reviewer 4 Report

Comments and Suggestions for Authors

Title

Communities of Uropodina (Acari: Mesostigmata) in nest boxes inhabited by dormice (Glis glis and Muscardinus avellanarius) and differences in percentage of nidicoles in nest of various hosts

by

Jerzy Błoszyk, Grzegorz Hebda, Marta Kulczak, Michał Zacharyasiewicz, Tomasz Rutkowski and Agnieszka Napierała

The submitted paper contains an analysis of the community of Uropodina (Acari: Mesostigmata), that inhabit nest boxes. In the examined boxes, the Authors stated the presence of five species belonging to the Uropodina group. It is interesting, that only one, i.e. Leiodinychus orbicularis, is a nidicole, i.e., a species inhabiting nests of various animals. In the reviewed article the habitat preferences of the mentioned species and other Uropodina are presented as well. The material for the study was collected from nest boxes in six forest complexes in south and south-western Poland. During the research the Authors have revealed the presence of five species of Uropodina.

The paper is written in a standard array, materials and methods are described adequately, the tables are informative. In my opinion, the reviewed paper lacks a map of the research area and photographs of the nest boxes used in the research.

To conclude, I am of an opinion that the article fits into the scope of “Animals” and could be published, but after a minor revision marked in the enclosed text has been done.

Author Response

Reviewer #4

Communities of Uropodina (Acari: Mesostigmata) in nest boxes inhabited by dormice (Glis glis and Muscardinus avellanarius) and differences in percentage of nidicoles in nest of various hosts

The authors of the study are grateful to the reviewer for all comments and suggestions. Most of them have turned out to be extremely helpful, which obviously has considerably improved the overall quality of the manuscript. The more detailed comments to the remarks are as follows: 

Detailed responses to the Reviewers comments: 

Response to Reviewer #4:

We’re grateful for the advice and for all the comments included in the text - we have taken all of them into account and it has certainly improved the quality of the manuscript. The map and photos of boxes have been added in the Methods section.